# Moderately Low Effectiveness of the Influenza Quadrivalent Vaccine: Potential Mismatch between Circulating Strains and Vaccine Strains

**DOI:** 10.3390/vaccines11061050

**Published:** 2023-05-31

**Authors:** Maaweya E. Awadalla, Haitham Alkadi, Modhi Alarjani, Abdullah E. Al-Anazi, Mohanad A. Ibrahim, Thamer Ahmad ALOhali, Mushira Enani, Wael Alturaiki, Bandar Alosaimi

**Affiliations:** 1Research Center, King Fahad Medical City, Riyadh Second Health Cluster, Riyadh 11525, Saudi Arabia; 2Comprehensive Cancer Center, King Fahad Medical City, Riyadh Second Health Cluster, Riyadh 11525, Saudi Arabia; 3Data Science Program, King Abdullah International Medical Research Center, Riyadh 11481, Saudi Arabia; 4Medical Protocol Department, Kind Abdulaziz Medical City, Ministry of National Guard Health Affairs, Riyadh 11426, Saudi Arabia; 5Dr. Sulaiman Alhabib Medical Group, Department of Medicine, Olaya Medical Complex, Riyadh 11643, Saudi Arabia; 6Department of Medical Laboratory Sciences, College of Applied Medical Sciences, Majmaah University, Majmaah 11952, Saudi Arabia

**Keywords:** influenza viruses, influenza A genotypes, vaccines, quadrivalent, effectiveness, phylogenetic analysis

## Abstract

The annual seasonal influenza vaccination is the most effective way of preventing influenza illness and hospitalization. However, the effectiveness of influenza vaccines has always been controversial. Therefore, we investigated the ability of the quadrivalent influenza vaccine to induce effective protection. Here we report strain-specific influenza vaccine effectiveness (VE) against laboratory-confirmed influenza cases during the 2019/2020 season, characterized by the co-circulation of four different influenza strains. During 2019–2020, 778 influenza-like illness (ILI) samples were collected from 302 (39%) vaccinated ILI patients and 476 (61%) unvaccinated ILI patients in Riyadh, Saudi Arabia. VE was found to be 28% and 22% for influenza A and B, respectively. VE for preventing A(H3N2) and A(H1N1)pdm09 illness was 37.4% (95% CI: 43.7–54.3) and 39.2% (95% CI: 21.1–28.9), respectively. The VE for preventing influenza B Victoria lineage illness was 71.7% (95% CI: −0.9–3), while the VE for the Yamagata lineage could not be estimated due to the limited number of positive cases. The overall vaccine effectiveness was moderately low at 39.7%. Phylogenetic analysis revealed that most of the Flu A genotypes in our dataset clustered together, indicating their close genetic relatedness. In the post-COVID-19 pandemic, flu B-positive cases have reached three-quarters of the total number of influenza-positive cases, indicating a nationwide flu B surge. The reasons for this phenomenon, if related to the quadrivalent flu VE, need to be explored. Annual monitoring and genetic characterization of circulating influenza viruses are important to support Influenza surveillance systems and to improve influenza vaccine effectiveness.

## 1. Introduction

Influenza viruses are highly contagious pathogens and have been observed as the most important medically attended acute febrile respiratory disease in humans. It is associated with a significant disease burden and increased costs for healthcare providers [1,2]. Worldwide, influenza viruses cause serious pandemics where millions of cases were reported with high mortality rates [3,4]. There are several influenza A virus subtypes that have been recorded throughout history to cause devastating pandemics to humans [2,4]. The ability of influenza A viruses to evolve and to cross species barriers resulted in the emergence of new subtypes such as; H5N1, H7N9, H9N2, H5N8, and H7N7 subtype [5,6,7,8,9].

Influenza A virus evolution, is controlled by several factors such as the origin and evolution of the hemagglutinin (HA) gene, receptor specificity, antigenic drift and shift, recombination, and crossing species barriers. Among these, antigenic drift and antigenic shift are the major mechanisms for virus evolution [10,11,12,13]. Antigenic drift and shift of the circulating Influenza strains result in divergent severity of influenza infection and become a continuous concern for public health [4,14]. The evaluation of influenza vaccine effectiveness (VE) is essential to illustrate how feasibility assessments support the national influenza vaccination programs and to report recommendations on circulating influenza strains for vaccine strain selection. Additionally, annual VE evaluations are essential to guide health authorities to introduce additional preventive methods [12,13] if the VE is low. Evolving knowledge on the development and fitness of antigenically drifted influenza A virus, changes in prevalence, and characterization of possible vaccine virus candidates are all variables that influence strain selection and vaccine component updates.

Saudi Arabia hosts religious mass gatherings during the Hajj and Umrah seasons when millions of people from highly diverse geographical regions visit the holy places in Mecca and Madinah. It has been estimated that roughly 24,000 individuals acquire influenza during each Hajj period, and pilgrims can import influenza infection back to their countries [15,16,17]. Such events urge researchers to study influenza vaccine effectiveness and to describe the molecular characterization of circulating influenza viruses in Saudi Arabia. The evaluation of influenza quadrivalent vaccine effectiveness (VE) is essential to illustrate how feasibility assessments support the national influenza vaccination programs, predict the next Influenza pandemic, and report recommendations on circulating influenza strains for vaccine strains selection. Annual VE evaluations are essential to sustain public confidence in influenza vaccination programs and to guide health authorities to introduce additional preventive methods if the VE is low. Therefore, the aim of this study was to evaluate and assess the quadrivalent influenza vaccine’s clinical effectiveness as well as the molecular characterization and phylogenetic analyses of circulating influenza strains (influenza A/H1N1pdm09 and A/H3N2).

## 2. Materials and Methods

### 2.1. Study Setting, Patients and Sample Collection

This single-center prospective study was conducted at King Fahad Medical City (KFMC), Riyadh, Saudi Arabia. The ILI was defined according to the WHO global influenza surveillance standards definitions for ILI and severe acute respiratory infections (SARI). All cases of individuals who visited the outpatient clinic with ILI signs and symptoms were included in this study. A total of 778 respiratory specimens (nasopharyngeal swab and/or mid-turbinate nasal swab) were prospectively collected and mixed with viral transport medium (Virocult kit) and kept at −80 °C until further use. Specimens were tested using real-time reverse transcription polymerase chain reaction (RT-PCR). We excluded patients <6 months of age, those who received an influenza vaccine less than two weeks before symptom or disease onset, those who had already taken anti-influenza viral drugs before the day of sample collection and those unwilling to participate. The research was carried out during the pre-COVID-19 pandemic and the early stages of the COVID-19 pandemic; all patients were enrolled during the period of influenza season (2019–2020). The total number of lifetime doses of the vaccine was also investigated. Each ILI patient completed a baseline questionnaire at the time of sample collection. Information on participant’s influenza vaccine status, age, sex, use of anti-influenza viral drugs or other immunosuppressants, chemo and/or physiotherapy, and history of autoimmune diseases, immunodeficiency, and chronic diseases.

### 2.2. Ethical Approval

The King Fahad Medical City Institutional Review Board (IRB) reviewed and approved the research protocol (IRB log number: 19-477). All work was conducted according to the principles of the Declaration of Helsinki. All participants provided written and signed informed consent.

### 2.3. Identification of Case and Control Patient

A test-negative design (TND) case-control study was used to estimate IVE. The TND provides reliable IVE estimates because it overcomes and reduces biases attributable to infection misclassification and due to differences in healthcare-seeking behavior among vaccinated and unvaccinated individuals as well as cases and controls. Unlike conventional case-control studies, the TND does not require health control groups (non-disease controls); in this study, clinical samples were collected from influenza-like illness (ILI) patients and categorized into cases and controls according to the influenza test results. Cases were confirmed and defined as an ILI patient with a positive influenza A or B virus by Real-time polymerase chain reaction (RT-PCR). Controls were ILI patients with negative RT-PCR results for any influenza viruses. We consider ILI patients vaccinated if they had received at least one dose of the quadrivalent 2019/2020 seasonal influenza vaccine more than two weeks before ILI onset. Research staff via medical interviews, obtained demographic information, patients’ symptoms, date of onset, illness characteristics, influenza vaccination status, vaccine type, date of vaccination, sex, and age (at the date of sample collection was stratified as 2–18, 19–55 and >55 years) and comorbidities and/or chronic disease.

### 2.4. Influenza Vaccination Status

In Saudi Arabia, all individuals aged ≥60 years old are recommended by the Ministry of Health to receive the seasonal influenza vaccine. The (INFLUVAC^®^ TETRA) quadrivalent inactivated influenza vaccine (2019–2020 seasons) contains A/Michigan/45/2015 (H1N1) pdm09-like virus, A/Switzerland/8060/2017 (H3N2)-like virus, B/Colorado/06/2017-like virus (B/Victoria/2/87 lineage) and B/Phuket/3073/2013-like virus (B/Yamagata/16/88 lineage) manufactured by Abbott Company (Chicago, IL, USA) is recommend recommended by the Ministry of Health in Saudi Arabia. Influenza vaccination status was collected and confirmed during medical interviews. All vaccinated participants in this study were vaccinated with the quadrivalent influenza vaccine (2019–2020 seasons). ILI patients were considered vaccinated for influenza if they received at least one dose of the influenza vaccine in the 12 months before the hospital visit for medical care. Patients who received the influenza vaccine less than two weeks before ILI symptom onset were excluded from our primary analysis.

### 2.5. Evaluation of VE

Influenza VE was calculated as [1-odds ratio (OR)] × 100%, and OR was calculated using the following equation:

(Number of influenza positives among vaccinated ILI patients X number of influenza negatives among unvaccinated ILI patients)/(Number of influenza negatives among vaccinated ILI patients X number of influenza positives among unvaccinated ILI patients).

VE estimates were calculated against any type of influenzas and separately for influenza A subtypes and B and lineages.

### 2.6. RNA Extraction

Viral RNA was extracted and purified from nasopharyngeal and pharyngeal swabs (140 ul) using the QIAamp Viral RNA extraction Mini kit (Qiagen, Hilden, Germany) following the manufacturer′s guidelines. Sterile RNase-free water was used instead of the specimen as a negative control. The concentration and purity of the extracted viral RNA were confirmed with NanoDrop 2000/2000c Spectrophotometer (Thermo Scientific, Waltham, MA, USA). Viral RNA purity was read at 260/280 nm, and concentrations were expressed in ng/μL. The purified viral RNA was eluted in 60 µL of elution buffer, frozen, and stored at −80 °C.

### 2.7. Molecular Detection Influenza A Viruses by One-Step Real-Time RT-PCR

Viral RNA from all samples was tested by RT-PCR assay using the Superscript III Platinum One-step qRT-PCR kit (Invitrogen, Waltham, MA, USA). The RT-PCR detection reaction (total volume 25 μL) contained 12.5 μL 2× reaction mix, 1 μL SuperScript III RT/Platinum One-step qRT-PCR mix, 0.5 μL (5 μM) probe and 1.5 μL (10 μM) of each one of the primers (sense primer and anti-sense) for the influenza A virus matrix (M) gene, 0.1 μL RNase inhibitor (10 U), 5 μL of extracted viral RNA and 2.9 μL Nuclease free water. The following thermal program in an ABI-7500 Fast Real-Time (RT-) PCR System was used: one cycle of reverse transcription and pre-denaturation for 30 min at 50 °C, 2 min at 95 °C for inactivation of cDNA synthesis and DNA polymerase activation followed by 45 amplification cycles of 95 °C for 15 s, 30 s at 56 °C and 15 s at 72 °C (extension step). The negative and positive controls were included in each experiment. The Fluorescence data were collected at the annealing step (56 °C) of each cycle. Each sample’s threshold cycle (Ct) was calculated by determining the fluorescence exceeding the threshold limit point. The positivity for influenza A was determined if the sample had a cycle-threshold (CT) value ≤ 45 and an exponential amplification result.

### 2.8. Molecular Subtyping of Influenza A and B Viruses

Samples tested positive for influenza type A form screening one-step RT-PCR was subsequently subtyped in a separate RT-PCR assay using PowerChek™ Influenza A&B Multiplex Real-time PCR Kit (PowerChek; Kogene Biotech, Seoul, Republic of Korea) following the manufacturer′s protocol, on the ABI-7500 Fast Real-Time (RT-) PCR System.

### 2.9. Molecular Phylogenetic Analysis of Influenza A (H3N2) and (H1N1) Viruses and Analysis of the Sequencing Data, Phylogenetic Tree

For full haemagglutinin (HA) and neuraminidase (NA) gene sequencing, all H3N2 and H1N1positive cases with high viral load (cycle threshold (Ct) ≤ 25) were selected and subjected to a specific one-step RT-PCR amplification protocol using (HA) and (NA) gene-specific primers according to the WHO information for the molecular detection of influenza viruses Global Influenza Programme [18]. The resulting amplicon of each (HA) and (NA) gene fragment were sequenced by the Sanger method [19] with the BigDye Direct Cycle Sequencing Kit in an ABI 3730xl DNA Sequencer (Applied Biosystems, Life Technologies, Foster City, CA, USA) at the Research laboratories, biomedical administration, Research Center, King Fahad Medical City, Saudi Arabia. Using Genius Tool, the collected sequences were subjected to quality control, including primer trimming and eliminating low-quality sequences. By submitting the sequences to a BLASTn search on NCBI, the identities of the sequences were confirmed, and reference sequences were acquired [20]. The gene segments were then extracted and concatenated with their respective GenBank gene references. Multiple sequence alignment was conducted using MAFFT with default settings [21,22]. A Bayesian phylogenetic study employing BEAUti and BEAST with a chain length of 100 million and the GTR substitution model was undertaken [23,24]. FigTree software was used to display the generated trees.

### 2.10. Statistical Analysis

Statistical analyses, calculations, and graphs were performed using GraphPad Prism software version 8.0 for Mac (GraphPad Software, San Diego, CA, USA). The primary analysis was the evaluation of VE against the influenza-associated illness for any influenza virus and influenza A subtypes and influenza B lineages. We compared characteristics among influenza-positive case patients and influenza-negative control patients using descriptive statistics and the student *t*-test. The test-negative design was used to estimate the ratio of the odds of vaccination among ILI patients testing positive for influenza A and/or influenza B to the odds of vaccination among ILI patients testing negative. VE was calculated as (1-aOR) × 100. The odds ratio was calculated using clinical variables. Phylogenetic trees were constructed using maximum likelihood methods and the best-fitting nt substitution model. *p* < 0.05 was considered statistically significant for all analyses.

## 3. Results

### 3.1. Characteristics of Study Participants

During the study period 2019/2020, 778 samples from ILI patients were collected and investigated by molecular methods. The median age of ILI patients was 31 years. Of 778 ILI patients, 265 (34%) were male, and 513 were female (66%). Approximately one-half (39%, *n* = 302) of ILI patients were vaccinated, 476 (61%) were not vaccinated in the study season, and 178 (23.1%) were vaccinated in the previous season. The number of laboratory-confirmed influenza cases was 171 (22%), including 151 (88%) cases of influenza A and 20 (12%) cases of influenza B, and there were 607 (78%) laboratory-influenza negative (see Table 1). The vaccine coverage among influenza-positive case-study patients was 29.8% compared to 41.3% in influenza-negative control patients. In this study, laboratory-confirmed influenza cases were slightly younger than ILI patients (controls) who tested negative for influenza (median age 30 vs. 32 years).

### 3.2. Pathogen Spectrum of Influenza Cases and Virological Description in 2019/2020

During the study period, 778 outpatients with ILI symptoms met the inclusion criteria and were included in the analysis. Among them, 171 (22%) were laboratory-confirmed positive for influenza virus infection, with 151 (88%) positive for influenza A viruses and 20 (12%) positive for influenza B viruses. Among influenza A virus infections, 91 (11.7%) were A (H3N2), 43 (5.5%) were A (H1N1) pdm09, 7 (0.9%) were co-infection with both H3N2 and H1N1 and 10 (1.2%) were unsubtyped influenza A. Among laboratory-confirmed influenza B positive cases, 2 (0.3%) were Yamagata-lineage, 12 (2%) were Victoria lineage, and 6 (0.9%) were unknown lineages. The virological description throughout the study season revealed that both A(H1N) and A(H3N2) detections overlapped. The total number of flu cases in Saudi between 2017 and 2022 provided the distribution of influenza subtypes and lineages (see Figure 1).

Surveillance of Influenza in Saudi Arabia revealed overlapping with the winter peak of influenza and other respiratory illnesses. We observed a one-fold decrease in the overall incidence of ILI cases during the 2020–2021 influenza season (6.2%) compared to the ILI cases during the 2019–2020 influenza season (13.1%). It indicates the need to differentiate between influenza-related infections and ILI cases of other respiratory infections such as SARS-CoV-2. This might be attributed to the fact that the population is practicing personal behavior changes adopted from the introduction of tight interventions during COVID-19 control measures such as masking and social distancing. After the COVID-19 pandemic, through 2021–2022, we noticed that the Flu A positivity rate started to decrease, and a Flu B wave started to increase. Flu B positive cases have become three-quarters of the total influenza positive cases in October-November 2022, indicating a nationwide flu B surge. The reasons for this phenomenon need to be explored.

### 3.3. Influenza A Subtypes Predominant during the Study Period (2019–2022 Seasons)

171 of 778 samples obtained from ILI patients (22%) were positive for influenza viruses. Data analysis revealed that the predominant subtype was influenza A (H3N2) (11.7%) in 2019/2020. Influenza A (H1N1) pdm09 (5.5%) and influenza B (3.3%) were co-circulated through the study period, similar to what has been observed in other countries [25,26]. However, there were only 2 cases of B/Yamagata lineage viruses circulating (see Figure 2).

### 3.4. Vaccine Effectiveness (VE) for Preventing Influenza Illness

We estimated influenza VE using a test-negative study design, comparing the odds of vaccination among case patients who tested positive for influenza with control patients who tested negative for influenza. The VE against any type of laboratory-confirmed influenza (A+B) among all ages and both genders was 28% (95% confidence interval [CI]: 17–38, and 22% (95% CI: 7–34) for influenza B. Among all ages in the target group for vaccination, the VE was 22% (95% CI: 7–34) in 2016–17 and 13% (95% CI: −21 to 38) in 2017–18. The results showed that the 2019–2020 seasonal influenza vaccine effectiveness was moderately low (39.7%) against laboratory-confirmed influenza. In order to estimate influenza vaccine effectiveness according to the virus subtype, we stratified the influenza laboratory-confirmed cases into A(H3N2) and A(H1N1) pdm09 virus subtype and influenza B (B/Victoria), and vaccine effectiveness was calculated separately for influenza A subtypes (H3N2 and H1N1) and influenza B and lineage (B/Victoria).

### 3.5. VE against Influenza A Subtypes

In order to estimate influenza vaccine effectiveness according the virus subtype, we stratified the influenza A laboratory confirmed into A(H3N2) and A(H1N1) pdm09 virus subtype, adjusted VE for preventing influenza A(H3N2) illness was 37.4% (95% CI: 43.7–54.3) and 39.2% (95% CI: 21.1–28.9) for preventing influenza A(H1N1) pdm09 illness.

### 3.6. Vaccine Effectiveness for Preventing Influenza B Illness

The overall VE for preventing influenza B illness was 39.2% (95% CI: 95% CI: 7.5–12.5). When we stratified influenza B by lineages, the adjusted VE for preventing influenza B, Influenza B Victoria lineage (B/Victoria), illness was 71.7% (95% CI: −0.9–3). VE for influenza B Yamagata lineage (B/Yamagata was not estimated as a consequence of the very limited number of positive cases (*n* = 2).

### 3.7. Vaccine Effectiveness according to Age Group

VE was sub-analyzed after laboratory-confirmed influenza cases were categorized into three age groups (2–18 years: group 1, 19–55 years: group 2, and >55 years: group 3). VE preventing influenza illness was estimated at 38.5% (95% CI: 5.5–1.5) in group 1 and 35% (95% CI: 58.8–70.2) in group 2. VE for group 3 was not calculated due to the limited number of positive cases (*n* = 5). VE varied by age group; the VE estimates were higher among those aged 2–18 years than those aged 19–55.

### 3.8. Vaccine Effectiveness according to Gender

The VE against any type of laboratory-confirmed influenza (A+B) among males was 50.11% (95% CI: 24.6–34.4) and 50.6% (95% CI: 7–34) for preventing influenza A illness, respectively. Among females, VE against any type of laboratory-confirmed influenza (A+B) was 33.4% (95% CI: 50.5–61.5) and 33.2% (95% CI: 43.8–54.2) for preventing influenza A illness, respectively.

### 3.9. Association of Prior Season (2018–2019) Vaccination on Current Season (2019–2020) VE

To further examine the effect of prior season vaccination on the study years VE, we stratified the ILI patients by prior season Influenza vaccination status (categorized as unvaccinated or received seasonal influenza vaccine in the previous influenza season). Patients were excluded from this analysis if prior season vaccination data were unavailable.

### 3.10. Phylogenetic Analysis

The low vaccine effectiveness revealed in this study encouraged us to investigate the possibility of a mismatch between the circulating and vaccine strains. We sequenced the complete gene (A/H3, A/H1, A/N2, A/N1) of H1N1 and H3N2 viruses. The results of our phylogenetic analysis showed that the samples in our dataset were distributed across various clades, indicating a genetic diversity of H1N1 and H3N2 viruses. Notably, one sample of H1N1 clustered with the A/Michigan/45/2015 (H1N1) pdm09-like virus, which was included in the vaccine to protect against the 2009 H1N1 pandemic strain (Figure 3). Similarly, H3N2 HA complete sequences clustered together in a single clade that closely related to the vaccine strain of A/Switzerland/8060/2017 (H3N2)-like virus suggesting their close genetic relatedness and indicating genetic similarities with circulating strains of H3N2-H3 in our dataset (Figure 4). Neuraminidase (NA) segment sequences showed genetic similarities to the vaccine strains (Appendix A). The nucleotide sequences and genomic data identified in this study have been submitted to the GenBank database; 24 complete A/H3 sequences under accession numbers OQ259932-OQ259955, 20 complete A/N2 sequences under accession numbers OQ263178-OQ263196, 14 complete A/H1 sequences under accession numbers OQ256175-OQ256188, and15 complete A/N1 sequences under accession numbers OQ256192-OQ256206.

## 4. Discussion

In this study, we performed a test-negative design method during the 2019–2020 influenza season at King Fahad Medical City (KFMC), Riyadh, Saudi Arabia. Annual seasonal influenza vaccination is the most effective way of preventing influenza illness and hospitalization. The quadrivalent influenza vaccine (2019–2020 seasons) contains A/Michigan/45/2015 (H1N1) pdm09-like virus, A/Switzerland/8060/2017 (H3N2)-like virus, B/Colorado/06/2017-like virus (B/Victoria/2/87 lineage) and B/Phuket/3073/2013-like virus (B/Yamagata/16/88 lineage). In Saudi Arabia, fewer published papers and national data are available regarding seasonal influenza vaccine effectiveness, specifically against high-risk groups for influenza-associated hospitalizations and emergency department visits limiting the data about VE profile. Saudi Arabia, unlike other countries in the world every year faces an exclusive and challenging situation of influenza outbreaks due to the Hajj and Umrah seasons, which may introduce the influenza A virus and new genotypes to the local circulating influenza A virus. Such challenges mean that the effectiveness of the seasonal influenza vaccine must be evaluated each year. Close monitoring of the circulating influenza A virus is crucial, and liaising with public and private Saudi health authorities to implement extraordinary public health responsibilities are necessary. During the Hajj season, approximately three million pilgrims are gathered from about 180 countries worldwide in the Kingdom of Saudi Arabia at the holy places in the cities of Makkah and Madinah [25,26,27]. The Hajj and Umrah assembly could be a vehicle that facilitates the mixing of the genetic pool of influenza A and other respiratory viruses and the generation of mutant variants that could affect the vaccine efficacy and be globally transmitted upon the return of pilgrims to their home countries [27,28]. The same risk continues throughout the year, involving over 10 million participants, and could provide a means of renewing the pool of circulating influenza and other respiratory viruses each year.

Approximately, one-half of ILI patients were vaccinated in the study season, and (23.1%) were vaccinated in the previous season. Previous data showed that the influenza vaccination rate among health workers was low in Saudi Arabia, and the unvaccinated group had a significantly higher rate of ILI as compared with the vaccinated group [29,30,31]. The Saudi Ministry of Health aims to provide vaccines for half a million people at a high risk of developing serious complications due to an infection of the seasonal influenza virus. They are all health care workers, pregnant women, immune deficiency patients congenital or acquired, children aged 6 months to 5 years, elderly, and patients with chronic illnesses. In Saudi Arabia, a commercial influenza vaccine is licensed. The Saudi National Program for Immunization aims to achieve an influenza vaccination rate of 75 percent of the high-risk population.

The number of laboratory-confirmed influenza cases was 178 (23%), including 151 (19.4%) cases of influenza A and 27 (26%) cases of influenza B, and there were 600 (77%) laboratory-influenza controls. The vaccine coverage among influenza-positive case-study patients was 29.8% compared to 41.3% in influenza-negative control patients. During the study period of the 2018–2019 influenza season in Saudi Arabia, analysis of our data revealed that the predominant subtype was influenza A (H3N2) (11.7%) in 2019–2020, followed by influenza A (H1N1) pdm09 (5.5%). In addition to the A(H3N2) and A(H1N1), we showed that influenza B strains both (Yamagata-lineage and Victoria) lineages were co-circulated through the study period; similar to what has been observed in other countries [12,32]. There were, however, only two cases of B/Yamagata lineage viruses in circulation. The evasion of host immune response via rapid molecular evolution and accumulation of mutations in major surface viral glycoproteins of the influenza A viruses A(H3N2) and A(H1N1) is a challenge and one of the critical factors for seasonal influenza vaccine preparation. The emergence of antigenically drifted viruses requires updating vaccine components each year. Vaccine strain selection requires analysis of circulating viruses from several parts around the world to predict the predominant or emerging influenza subtypes [12,33]. The selection of a new vaccination reference virus may necessitate more preparation time, which may have public health implications if the seasonal influenza vaccine is delayed [32].

INFLUVAC^®^ TETRA has some disadvantages that should be taken into account, such as changes in the glycosylation patterns of the viral proteins and increased mutations in the HA that have a negative effect on antibody sites in the HA protein, as well as variability in vaccine efficacy. These changes can result in a reduction in vaccine efficacy against the circulating homologous virus, especially when there is a mismatch between the vaccine strain and the circulating strain. In recent years, there has been a growing interest in developing alternative technologies for influenza vaccine production that do not rely on eggs. One such approach is cell-based vaccine production. This approach has been shown to have several advantages over egg-based vaccines, including improved yields, faster production times, and a reduced risk of egg-related allergic reactions. It also does not acquire mutations in the HA protein, which keep the influenza viruses antigenically similar to the circulating strains.

In this study, we observed VE against any type of laboratory-confirmed influenza (A or B) among all ages and genders: 28% and 22% for influenza A and B, respectively. Our results showed that the 2019–2020 seasonal influenza vaccine effectiveness was moderately low (39.7%) against laboratory-confirmed influenza. Our VE estimates agree with VE estimates performed in 2018–2019 in Saudi Arabia [34]. Our data are consistent with several studies showing lower influenza VE (32–38%); for example, in Australia and Canada [35,36]. A moderate (59% to 69%) VE against influenza was reported in many countries [37,38]. However, a higher VE (52%) was estimated in other studies [39]. When we stratified the influenza A laboratory confirmed into A(H3N2) and A(H1N1)pdm09 virus subtype, adjusted VE for preventing influenza A(H3N2) illness was 37.4% and 39.2% for preventing influenza A(H1N1)pdm09 illness.

It is worth mentioning that we did not evaluate seasonal influenza vaccine effectiveness for the entire season due to the COVID-19 pandemic, and when the first COVID-19 was officially reported from the Ministry of Health, we decided to stop the collection of samples and ILI patient recruitment, and this seemed to play a significant role in VE. During the study period, we reported a modest VE for all influenza and the lowest for influenza A (H3N2), with a higher VE for influenza B, particularly in the early phase of the season.

In our study, only a few influenza cases were reported in the group with the eldest. It is possible that the few infections reported in the group with the eldest were due to a number of factors. First, the elderly participants in the study may have been taking other preventative measures, such as avoiding close contact with sick individuals, avoiding crowded public places, staying at home, or practicing good hand hygiene, which could have reduced their risk of infection. Second, this could be due to individual variability in immune function. Third, the relatively small sample size could be one reason. It is also important to note that this finding may not be generalizable to other populations or influenza seasons. Overall, it is important to consider all these factors when interpreting the study results and to recognize that the few infections reported in the group with the eldest may be due to a combination of factors rather than any single factor alone. Further research is needed to better understand the effectiveness of influenza vaccination in the elderly and to develop strategies to improve vaccine efficacy in these groups.

Experience from the 2019–2020 influenza season vaccine effectiveness study highlights recent challenges with VE against influenza A(H3N2) and A(H1N1)pdm09 viruses and the need for more accurate vaccine component selection. To increase VE improvement, vaccine approaches that provide wide protection against antigenically distinct groups of both A(H3N2) and A(H1N1) viruses are required if diverse antigenically shifted groups of viruses continue to co-circulate [40].

The findings of our study align with similar studies that have investigated the genetic diversity and relatedness of influenza virus strains [41,42]. Similarly, a study by [43] found that the majority of H3N2 viruses circulating during the 2012–2013 season in the United States were antigenically similar to the vaccine strain. However, as we observed in our study, the presence of divergent clades of H3N2 viruses, has also been documented [43,44]. In this study, we observed a homogeneous pattern of the A/H3 virus circulation, whereas A/H1 sequences clustered into different phylogenetic groups. The molecular characterization of the circulating influenza A/H3N2 and H1N1 viruses highlighted substantial diversity in the A/H1 sequences. Our laboratory previously demonstrated the multiple introductions and co-circulation of several clades of seasonal H1N1 influenza viruses and the predominance of clade 6B.1 in Saudi Arabia [45].

The sequence alignment did not show any indication of positive or negative selection. The phylogenetic analyses revealed that most of the circulating influenza A/H3N2 strains clustered with the A/Switzerland/8060/2017 (H3N2)-like virus, which was included in the 2019–2020 seasonal influenza vaccine strain [46,47]. Moreover, the A/H3 sequences of the influenza A virus subtype showed amino acid sequences identical to each other. Sequence analyses of influenza virus subtypes H3 and H1 performed in the current study indicated identity with influenza A (H1N1 and H3N2) variants circulating globally, suggesting the reintroduction and spread to Saudi Arabia through the mass gatherings of Hajj and Umrah. The Hajj and Umrah assembly could be a vehicle that facilitates the mixing of the genetic pools of influenza A and other respiratory viruses and the generation of mutant variants that could affect vaccine efficacy and globally transmit respiratory pathogens upon the return of pilgrims to their home countries [27,28]. The same risk continues throughout the year, involving over 10 million participants, and could provide a means of renewing the pool of circulating influenza and other respiratory viruses each year.

In summary, our estimates showed that the seasonal influenza vaccine offered moderate protection against laboratory-confirmed influenza infection in the 2019–2020 winter season. We plan to increase the size of our study on seasonal influenza vaccine effectiveness for the coming winter season by including different regions to increase the potential sample size and generate more reliable and representative VE estimates in Saudi Arabia, particularly in the early and late winter season. An early and reliable VE estimate would also inform health authorities to consider alternative prevention approaches if VE was discovered to be low.

## 5. Conclusions

### 5.1. Implication in Clinical Practice

Seasonal influenza vaccination is moderately effective against laboratory-confirmed influenza pneumonia in 2019–2020. Given the increasing burden of influenza virus infection complications such as pneumonia, particularly in high-risk groups, such challenges necessitate an annual evaluation of the seasonal influenza vaccine effectiveness and close monitoring of the circulating influenza A virus is of great significance and also confronts public and private Saudi health authorities with extraordinary public health responsibilities. The implementation of non-pharmaceutical interventions (NPIs) aimed at reducing the spread of COVID-19, such as mask-wearing, social distancing, and hand hygiene, during the COVID-19 pandemic has led to a general reduction in the incidence of influenza and other respiratory pathogen infections. In many countries around the world, there was a notable decrease in the number of cases of influenza during the COVID-19 pandemic. We believe that the COVID-19 pandemic has highlighted the importance of influenza vaccination, even when influenza virus circulation is reduced, as a strategy to prevent severe influenza illnesses and reduce the burden on healthcare systems. Data from this study may improve seasonal influenza vaccination coverage and inform policymakers on improvements in vaccine effectiveness.

### 5.2. Limitations of the Study

We did not evaluate seasonal influenza vaccine effectiveness for the entire season due to the COVID-19 pandemic, and when the first COVID-19 was officially reported from the Ministry of Health, we decided to stop the collection of samples and ILI patient recruitment, and this seemed to play a significant role in VE. Although the sample size was relatively small to estimate influenza B subtype-specific VE, it did not affect our IVE estimates. Besides, the TND has some limitations, such as selection bias, limited information on the vaccine, and the potential for misclassification bias. Finally, some limitations that can affect the generalizability of our results include the convenience sampling method, which may not represent the entire population, and the characteristics of the study participants, such as age and gender.

### 5.3. Recommendations for Future Research

We plan to increase the size of our study on seasonal influenza vaccine effectiveness for the coming winter season by including different regions to increase the potential sample size and generate more reliable and representative VE estimates in Saudi Arabia for the entire season. An early and reliable VE estimate would also inform health authorities to consider alternative prevention approaches if VE was discovered to be low. The yearly development of seasonal influenza vaccines is a challenging race against the clock to detect and identify the influenza strains that will be included in the vaccine component. Further analysis should include deep genetic characterization of circulating influenza viruses as well as unsubtyped influenza A viruses and unknown influenza B lineages. Careful monitoring of genetic changes in the A/H genes, particularly the HA1 domain, during seasonal influenza times, may help in the identification of the immune escape strains and offer reliable data on newly emerging strains. The presence of divergent clades of H3N2 viruses highlights the need for ongoing genetic analysis and surveillance of influenza viruses to inform vaccine development and support national efforts to improve influenza vaccine effectiveness and public health measures. Such viral genetic characterization and genomic data could be used in the development of an effective regional-based vaccine according to the circulating strains.

## Figures and Tables

**Figure 1 vaccines-11-01050-f001:**
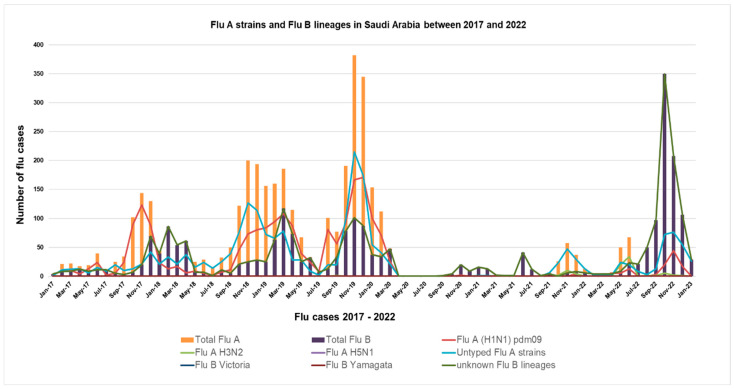
Distribution of samples positive for influenza by subtype and lineages between 2017 and 2022 according to [WHO EMRO]. https://www.emro.who.int/index.html (accessed on: 15 February 2023).

**Figure 2 vaccines-11-01050-f002:**
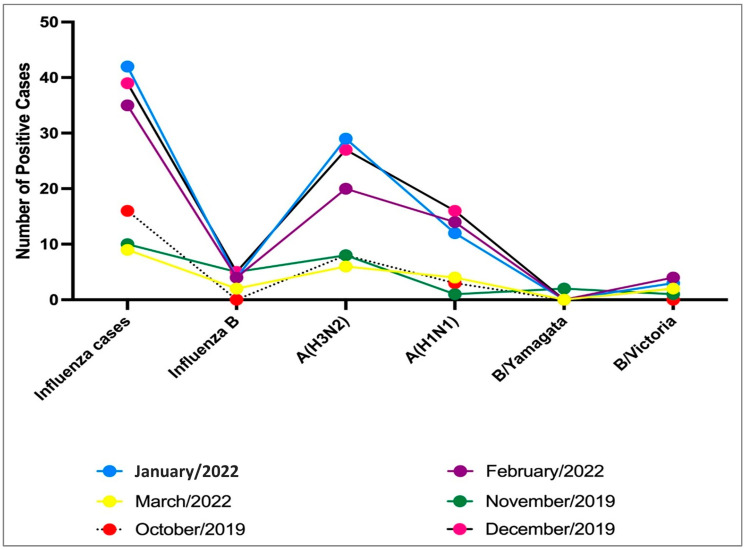
Number of ILI patients tested positive by months in the column chart (*n* = 171).

**Figure 3 vaccines-11-01050-f003:**
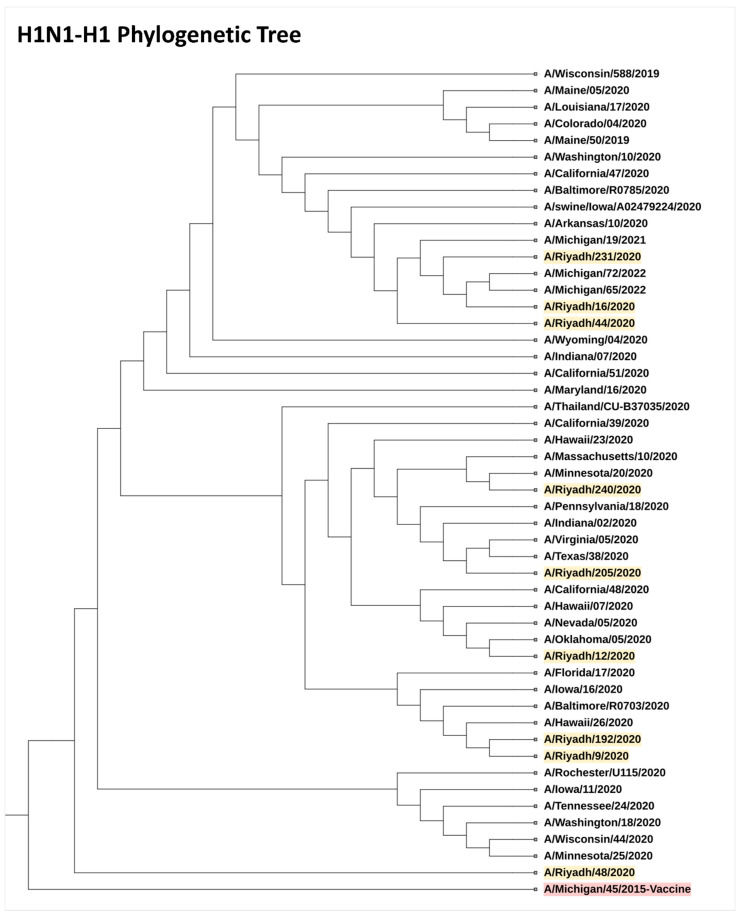
Phylogenetic tree of H1N1-Hemagglutinin (HA) gene showing the genetic diversity and evolutionary relationships among the analyzed samples (yellow highlight). Of note, one sample clustered with the A/Michigan/45/2015 vaccine strain (red highlight).

**Figure 4 vaccines-11-01050-f004:**
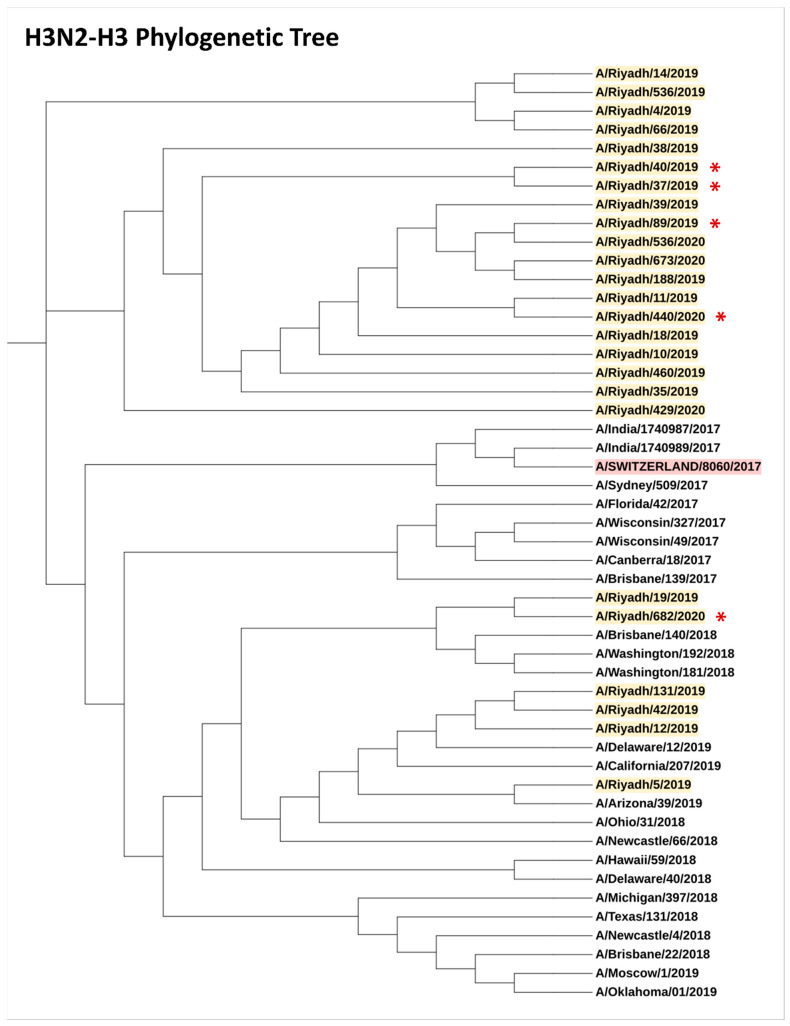
Phylogenetic tree of H3N2-Hemagglutinin (HA) segment showing the genetic diversity and evolutionary relationships among the analyzed samples (yellow highlight). Of note, 6 of our H3N2 HA complete sequences clustered together in a single clade that closely related to the vaccine strain of A/Switzerland/8060/2017 (red highlight). The starred strains (✱) are for vaccinated patients.

**Table 1 vaccines-11-01050-t001:** Demographic characteristics of ILI patients included in the study.

Baseline Variables	All Patients	Vaccinated	Unvaccinated	*p*-Value
	(*n* = 778)	*n* = 302 (39%)	*n* = 476 (61%)	
Characteristics:				
Age				
Median	31 ± 10	31 ± 7	31 ± 12	
Range	3–88	21–70	3–88	
Gender				
Men	265 (34%)	161 (61%)	104 (39%)	0.860 *
Women	513 (66%)	315 (61%)	198 (39%)	
Chronic Diseases				
Healthy	735 (94%)	282 (38%)	453 (62%)	0.286 *
Diseased	43 (6%)	20 (47%)	23 (53%)	
Flu infection	171 (22%)	51 (30%)	120 (70%)	
Flu A	151 (88%)	45 (30%)	106 (70%)	0.985 *
Flu B	20 (12%)	6 (30%)	14 (70%)	

* Not significant.

## Data Availability

The data generated and/or analyzed during the current study are available from the corresponding author on reasonable request.

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
