# Peer review of "Moderately Low Effectiveness of the Influenza Quadrivalent Vaccine: Potential Mismatch between Circulating Strains and Vaccine Strains"

_vaccines, 2023, doi:10.3390/vaccines11061050_

Round 1
Reviewer 1 Report
This nice manuscript by Awadalla et al. aims at evaluating and assessing the quadrivalent influenza vaccine's clinical effectiveness as well as the molecular characterization and phylogenetic analyses of circulating influenza strains (influenza A/H1N1pdm09 and A/H3N2).
Overall the manuscript is well written, and specifically the methods are well described and the results nicely reported.
I only have minor comments:
- line 352: you correctly state that calculating VE for B/Yamagata was not possible as only two cases were reported during 2019/20. Two important studies reported how influenza B/Yamagata appears to have disappeared after the emergence of COVID-19 (1-2). It would be interesting to mention these two studies and have a (cautious) comment on what could the implications be for the future of influenza vaccination.
- in the "limitations" paragraph, please report the limitations of using the test-negative design to calculate influenza vaccine effectiveness (3).
- considering the (rather usual) moderate effectiveness of influenza vaccination and the fact that the virus reduced its circulation during the past 2 seasons, increasing share of the population is now suspected to be susceptible for influenza infections (4,5). I suggest commenting this, mentioning what are the possible implications not only for future influenza epidemics, but also what are the potential challenges and opportunities for future larger studies aimed at evaluating influenza VE.
1. https://pubmed.ncbi.nlm.nih.gov/36177871/
2. https://pubmed.ncbi.nlm.nih.gov/36240828/
Moderate editing of English language is required.
Author Response
Response to Reviewers' comments
Dear Editor,
Thank you very much for the opportunity to review our manuscript. We would like to thank all the reviewers for their helpful comments and suggestions. We appreciate the comments and suggestions made by the reviewers. We strongly believe that the comments and suggestions have increased the scientific value of the revised manuscript by many folds. We have taken all the comments to improve and clarify the manuscript. We have considered all the changes suggested by the reviewers and they could be tracked as red color within the revised manuscript. Please see our point-by point specific responses (red color) to the reviewer’s comments below.
Reviewer 1
Comments and Suggestions for Authors
This nice manuscript by Awadalla et al. aims at evaluating and assessing the quadrivalent influenza vaccine's clinical effectiveness as well as the molecular characterization and phylogenetic analyses of circulating influenza strains (influenza A/H1N1pdm09 and A/H3N2).
Overall the manuscript is well written, and specifically the methods are well described and the results nicely reported.
I only have minor comments:
Response: We thank the reviewer 1 for providing positive words and comments, which we appreciate.
- line 352: you correctly state that calculating VE for B/Yamagata was not possible as only two cases were reported during 2019/20. Two important studies reported how influenza B/Yamagata appears to have disappeared after the emergence of COVID-19 (1-2). It would be interesting to mention these two studies and have a (cautious) comment on what could the implications be for the future of influenza vaccination.
Response: Thank for you providing these studies. The studies provided by reviewer have been cited. The selection of strains for influenza vaccines was challenging due to the unpredictable circulation of the influenza B/Victoria and B/Yamagata lineages, and influenza B lineage mismatches were frequent. Additionally, both lineages have evolved distinct antigens, thereby precluding any possibility of cross-protection. The potential disappearance or reduced detection of B/Yamagata could have significant implications for the use of QIV, which are currently used on a global scale. Several possible scenarios could be applied, including the continuation of QIV with a new strain that may help reduce vaccine mismatches, or a return to trivalent influenza vaccines to reduce immune and vaccine pressures that may lead to antigenic drift mutations.
- in the "limitations" paragraph, please report the limitations of using the test-negative design to calculate influenza vaccine effectiveness (3).
Response: Thank you for this comment. The test-negative design (TND) is commonly used in studies of influenza vaccine effectiveness. While the TND has some advantages over other study designs, it also has some limitations, such as selection bias, limited information on the vaccine, and the potential for misclassification bias. Despite these limitations, the TND can be a useful method for estimating influenza vaccine effectiveness, particularly in settings where other study designs are not feasible. However, the limitations should be carefully considered when interpreting the results. Minimizing selection bias in the TND requires careful study design, appropriate control group selection, standardized diagnostic criteria, adjustment for potential confounding factors, and sensitivity analyses. The limitations of test-negative design to calculate for influenza vaccine effectiveness have been added. See limitation section.
- considering the (rather usual) moderate effectiveness of influenza vaccination and the fact that the virus reduced its circulation during the past 2 seasons, increasing share of the population is now suspected to be susceptible for influenza infections (4,5). I suggest commenting this, mentioning what are the possible implications not only for future influenza epidemics, but also what are the potential challenges and opportunities for future larger studies aimed at evaluating influenza VE.
Response: Thank you for this important point, and we completely agree with you. The implementation of non-pharmaceutical interventions (NPIs) aimed at reducing the spread of COVID-19, such as mask-wearing, social distancing, and hand hygiene, during the COVID-19 pandemic have led to a general reduction in the incidence of influenza and other respiratory pathogen infections. The moderate effectiveness of influenza vaccination may increase the percentage of the population susceptible to influenza infections. This raises concerns for future influenza epidemics, as a larger group of susceptible individuals may lead to a more severe infection and a higher mortality rate. In addition, the increased susceptibility of the population presents both challenges and opportunities for future larger studies aimed at evaluating influenza vaccine effectiveness (VE). On the one hand, it may be more difficult to achieve high levels of VE due to the larger numbers of susceptible individuals. On the other hand, larger studies may be able to better assess the impact of vaccination on reducing influenza transmission and preventing severe disease in a different population. This statement has been mentioned in the conclusion section.
Again, we appreciate all of your insightful comments. Thank you for taking the time and energy to help us improve the manuscript.
Closing comments to the editor:
We thank the reviewers for the time they put into reviewing our paper, and we look forward to meeting their expectations. The comments and suggestions provided valuable insights and improved the manuscript. We would also like to explicitly acknowledge their contribution.
Once again, thank you for receiving our manuscript and considering it for publication in your esteemed journal. We hope our revisions will meet with your approval
Reviewer 2 Report
Overall, an interesting paper. I have only a few comments.
Viral circulation (severity and sub-strains); herd immunity/vaccination coverage rate (VCR), travel patterns, and other relevant effect modifiers are highly dependent on geographic region. I would advise to include the study location (Riyadh, Saudi Arabia) in the title and in the abstract. Now it takes too long before the reader realizes where the study was done.
Results are confusing. How can VE of the two strains (28% for A and 22% for B) be lower than the overall of 39.7%? Same confusion of the sub-strains: How can VE of the two A subtypes (37.4% for H3N2 and 39.2% for H1N1) be higher than the overall VE of 28% for A? I personally think this is caused by the choice to use non-collapsible Odds Ratios (OR), a serious limitation of the TND in my opinion.
It is not clear which measured confounders were included in the logistic regression to calculate the OR. I assume all the variables collected in the questionnaire, but it is important to mention this in the Methods.
line 83: "All cases of individuals who were visited the outpatient clinic with ILI signs and symptoms were included in this study". There is no mentioning of obtaining informed consent (just the refusal to participate in the study). That is unusual for a prospective study. Please explain why consent and IRB approval was not needed. In addition, I would like to know how many people refused to participate in the study as this might be an indication of selection bias.
line 84: typo: "A778 respiratory specimens"
The authors generalize the study results to entire Saudi Arabia. Please explain why a 778-subject convenience sample from Riyadh is representative to entire Saudi Arabia. If there are limitations to this generalization, please list these in the Discussion.
The authors imply that TND produces unbiased estimates "provides reliable IVE estimates", and there are no limitations (of TND) listed in the Discussion. I must agree that the most common pitfalls of TND (e.g., using rapid tests) were avoided, and that TND is probably one of the better methods to estimate VE. However, this doesn't mean that TND is without limitations. Mentioning a few limitations of TND will make the paper stronger. For examples see:
1. Foppa IM, Haber M, Ferdinands JM, et al. The case test-negative design for studies of the effectiveness of influenza vaccine. Vaccine. 2013;31(30):3104–3109.
2. Jackson ML, Nelson JC. The test-negative design for estimating influenza vaccine effectiveness. Vaccine. 2013;31(17): 2165–2168.
3. De Serres G, Skowronski DM, Wu XW, et al. The test-negative design: validity, accuracy and precision of vaccine efficacy estimates compared to the gold standard of randomised placebo-controlled clinical trials. Euro Surveill. 2013;18(37): 20585.
4. Sullivan SG, Feng S, Cowling BJ. Potential of the test-negative design for measuring influenza vaccine effectiveness: a systematic review
5. https://doi.org/10.1093/aje/kww063
Author Response
Response to Reviewers' comments
Dear Editor,
Thank you very much for the opportunity to review our manuscript. We would like to thank all the reviewers for their helpful comments and suggestions. We appreciate the comments and suggestions made by the reviewers. We strongly believe that the comments and suggestions have increased the scientific value of the revised manuscript by many folds. We have taken all the comments to improve and clarify the manuscript. We have considered all the changes suggested by the reviewers and they could be tracked as red color within the revised manuscript. Please see our point-by point specific responses (red color) to the reviewer’s comments below.
Reviewer 2
Comments and Suggestions for Authors
Overall, an interesting paper. I have only a few comments.
Response: We thank you for his comments and insights, which helped us strengthen our manuscript.
Viral circulation (severity and sub-strains); herd immunity/vaccination coverage rate (VCR), travel patterns, and other relevant effect modifiers are highly dependent on geographic region. I would advise to include the study location (Riyadh, Saudi Arabia) in the title and in the abstract. Now it takes too long before the reader realizes where the study was done.
Response: We wish to thank the reviewer for bringing this point to our attention. The location of the study (Riyadh, Saudi Arabia) has been added to the abstract of the paper.
Results are confusing. How can VE of the two strains (28% for A and 22% for B) be lower than the overall of 39.7%? Same confusion of the sub-strains: How can VE of the two A subtypes (37.4% for H3N2 and 39.2% for H1N1) be higher than the overall VE of 28% for A? I personally think this is caused by the choice to use non-collapsible Odds Ratios (OR), a serious limitation of the TND in my opinion.
Response: Thank you for this comment. We apologize for not making it clear. VE estimates were calculated against any type of influenza including influenza A unsubtyped and subtypes (H3N2 and H1N1) and influenza B (yamagata and Victoria) and unknown lineages. In order to estimate influenza vaccine effectiveness according the virus subtype, we stratified the influenza laboratory confirmed cases into A(H3N2) and A(H1N1) pdm09 virus subtype and influenza B (B/Victoria) and vaccine effectiveness was calculated separately for influenza A subtypes (H3N2 and H1N1) and influenza B and lineage (B/Victoria). Additionally, the limitations of test-negative design to calculate for influenza vaccine effectiveness have been added. See limitation section.
It is not clear which measured confounders were included in the logistic regression to calculate the OR. I assume all the variables collected in the questionnaire, but it is important to mention this in the Methods.
Response: Thank you for this comment. The related information has been added to the Materials and Methods section.
line 83: "All cases of individuals who were visited the outpatient clinic with ILI signs and symptoms were included in this study". There is no mentioning of obtaining informed consent (just the refusal to participate in the study). That is unusual for a prospective study. Please explain why consent and IRB approval was not needed. In addition, I would like to know how many people refused to participate in the study as this might be an indication of selection bias.
Response: We apologize for not mentioning this. We observed no participant refusal. “This study was reviewed and approved by the Institutional Review Board at King Fahad Medical City (IRB register number (19-477). All participants provided written and signed informed consent, and the study was carried out in accordance with the Helsinki Declaration”. This statement has been added to the Materials and Methods section. See subhead 2.2. Ethical Approval.
line 84: typo: "A778 respiratory specimens"
Response: Thank you for this comment. We realized that this was typo/mistake. This point has been corrected as “A total of 778 respiratory specimens”. See section 2.1.
The authors generalize the study results to entire Saudi Arabia. Please explain why a 778-subject convenience sample from Riyadh is representative to entire Saudi Arabia. If there are limitations to this generalization, please list these in the Discussion.
Response: Thank you for this comment. The limitations that can affect the generalizability of results have been added to the discussion section.” Some limitations that can affect the generalizability of our results include the convenience sampling method, which may not represent the entire population, and the characteristics of the study participants, such as age and gender”.
The authors imply that TND produces unbiased estimates "provides reliable IVE estimates", and there are no limitations (of TND) listed in the Discussion. I must agree that the most common pitfalls of TND (e.g., using rapid tests) were avoided, and that TND is probably one of the better methods to estimate VE. However, this doesn't mean that TND is without limitations. Mentioning a few limitations of TND will make the paper stronger. For examples see:
Response: Thank you for this important point and we completely agree with you. We apologize for not mentioning the limitations of TND. The TND has some limitations, such as selection bias, limited information on the vaccine, and the potential for misclassification bias. Despite these limitations, the TND can be a useful method for estimating influenza vaccine effectiveness, particularly in settings where other study designs are not possible. However, the limitations should be carefully considered when interpreting the results. Minimizing selection bias in the TND requires careful study design, appropriate control group selection, standardized diagnostic criteria, adjustment for potential confounding factors, and sensitivity analyses. As suggested, the limitations of test-negative design to calculate for influenza vaccine effectiveness have been added. See limitation section.
Again, we appreciate all of your insightful comments. Thank you for taking the time and energy to help us improve the manuscript.
Closing comments to the editor:
We thank the reviewers for the time they put into reviewing our paper, and we look forward to meeting their expectations. The comments and suggestions provided valuable insights and improved the manuscript. We would also like to explicitly acknowledge their contribution.
Once again, thank you for receiving our manuscript and considering it for publication in your esteemed journal. We hope our revisions will meet with your approval
Reviewer 3 Report
This article is very important both for science and for practical health care. However, some major and minor points need to be clarified.
The authors mentioned, “In Saudi Arabia, a commercial influenza vaccine is licensed” (see line 340). However, this information is insufficient. A wide range of readers, as well as vaccine experts, may not know what types of flu vaccines (whole virion inactivated, split, subunit, LAIV etc.) are licensed in any particular country. Saying that the vaccinees were immunized with the influenza vaccine is like saying that the grocery store sells cheese. Influenza vaccines are available in many varieties from different manufacturers, as are varieties of cheese.
To conclude on the effectiveness of the influenza vaccine, information should be provided on the type of vaccine – live or inactivated vaccine was used for vaccination. The manufacturer of the vaccine must also be listed in the "Materials and Methods". In accordance with WHO recommendations, vaccines may contain "–like viruses". Strains that are similar to those recommended may differ in immunogenicity from each other, which may also affect the results. “Discussion” and “Conclusions” should be made taking into account this information.
In addition, it was necessary to specify whether all of those observed during the entire observation period were vaccinated with the vaccine of the same type and the same manufacturer. If not, this should be noted in the “Limitations of the study”.
“Conclusion” contains only the most general words. Since the research was carried out in a very interesting period – during the pre–COVID–19 pandemic and its early stages, I would recommend adding to the “|Conclusion” the opinion of the authors on the effectiveness of influenza vaccination during the period when the circulation of influenza virus strains was significantly reduced and whether influenza vaccination is necessary in principle in a situation where there is little or no circulation of influenza viruses.
Author Response
Response to Reviewers' comments
Dear Editor,
Thank you very much for the opportunity to review our manuscript. We would like to thank all the reviewers for their helpful comments and suggestions. We appreciate the comments and suggestions made by the reviewers. We strongly believe that the comments and suggestions have increased the scientific value of the revised manuscript by many folds. We have taken all the comments to improve and clarify the manuscript. We have considered all the changes suggested by the reviewers and they could be tracked as red color within the revised manuscript. Please see our point-by point specific responses (red color) to the reviewer’s comments below.
Reviewer 3
Comments and Suggestions for Authors
This article is very important both for science and for practical health care. However, some major and minor points need to be clarified.
Response: We thank the reviewer 3 for reviewing our manuscript and providing positive words and comments, which we appreciate.
The authors mentioned, “In Saudi Arabia, a commercial influenza vaccine is licensed” (see line 340). However, this information is insufficient. A wide range of readers, as well as vaccine experts, may not know what types of flu vaccines (whole virion inactivated, split, subunit, LAIV etc.) are licensed in any particular country. Saying that the vaccinees were immunized with the influenza vaccine is like saying that the grocery store sells cheese. Influenza vaccines are available in many varieties from different manufacturers, as are varieties of cheese.
To conclude on the effectiveness of the influenza vaccine, information should be provided on the type of vaccine – live or inactivated vaccine was used for vaccination. The manufacturer of the vaccine must also be listed in the "Materials and Methods". In accordance with WHO recommendations, vaccines may contain "–like viruses". Strains that are similar to those recommended may differ in immunogenicity from each other, which may also affect the results. “Discussion” and “Conclusions” should be made taking into account this information.
In addition, it was necessary to specify whether all of those observed during the entire observation period were vaccinated with the vaccine of the same type and the same manufacturer. If not, this should be noted in the “Limitations of the study”.
Response: Thank you for this comment. The (INFLUVAC® TETRA) quadrivalent inactivated influenza vaccine (2019–2020 sea-sons) contains A/Michigan/45/2015 (H1N1) pdm09-like virus, A/Switzerland/8060/2017 (H3N2)-like virus, B/Colorado/06/2017-like virus (B/Victoria/2/87 lineage) and B/Phuket/3073/2013-like virus (B/Yamagata/16/88 lineage) manufactured by Abbott com-pany is recommend recommended by the Ministry of Health in Saudi Arabia. Influenza vaccination status were collected and confirmed during medical interviews. All vaccinat-ed participants in this study were vaccinated with the quadrivalent influenza vaccine (2019–2020 seasons). This statement has been added to the Materials and Methods section subhead 2.4.
“Conclusion” contains only the most general words. Since the research was carried out in a very interesting period – during the pre–COVID–19 pandemic and its early stages, I would recommend adding to the “|Conclusion” the opinion of the authors on the effectiveness of influenza vaccination during the period when the circulation of influenza virus strains was significantly reduced and whether influenza vaccination is necessary in principle in a situation where there is little or no circulation of influenza viruses.
Response: Thank you for this important point, and we completely agree with you. The implementation of non-pharmaceutical interventions (NPIs) aimed at reducing the spread of COVID-19, such as mask-wearing, social distancing, and hand hygiene, during the COVID-19 pandemic have led to a general reduction in the incidence of influenza and other respiratory pathogen infections. In many countries around the world, there was a notable decrease in the number of cases of influenza during the COVID-19 pandemic. Several factors may have led to the moderate effectiveness of influenza vaccination observed in our study. This highlights the potential benefits of NPIs in preventing the spread of respiratory infections. At the same time, there has been a decrease in the use of other vaccines, such as those for influenza, as people have been staying home more and taking other precautions to avoid exposure to respiratory viruses, especially during the pre–COVID–19 pandemic and its early stages. We do believe that the COVID-19 pandemic has highlighted the importance of influenza vaccination, even when the influenza viruses circulation is reduced, as a strategy to prevent severe influenza illnesses and reduce the burden on healthcare systems. This could have long-term implications for the occurrence of future influenza epidemics and the severity of infections. As suggested, we have added a statement showing our opinion to the conclusion in the discussion section.
Again, we appreciate all of your insightful comments. Thank you for taking the time and energy to help us improve the manuscript.
Closing comments to the editor:
We thank the reviewers for the time they put into reviewing our paper, and we look forward to meeting their expectations. The comments and suggestions provided valuable insights and improved the manuscript. We would also like to explicitly acknowledge their contribution.
Once again, thank you for receiving our manuscript and considering it for publication in your esteemed journal. We hope our revisions will meet with your approval